# Purification, Identification and Molecular Docking of Novel Antioxidant Peptides from Walnut (*Juglans regia* L.) Protein Hydrolysates

**DOI:** 10.3390/molecules27238423

**Published:** 2022-12-01

**Authors:** Luhao Fan, Xiaoying Mao, Qingzhi Wu

**Affiliations:** School of Food Science and Technology, Shihezi University, Shihezi 832003, China

**Keywords:** protein hydrolysate, structural characteristic, antioxidant activity, LC-MS/MS, molecular docking, structure-activity relationship

## Abstract

Walnut protein isolate (WPI) was hydrolyzed using Alcalase for 0, 30, 60, 90, 120 and 150 min to investigate the effect of different hydrolysis times on the structure and antioxidant properties of walnut proteins. The identified peptides HADMVFY, NHCQYYL, NLFHKRP and PSYQPTP were used to investigate the structure-activity relationship by using LC-MS/MS and molecular docking. The kinetic equations DH = 3.72ln [1 + (6.68 E_0_/S_0_ + 0.08) t] were developed and validated to explore the mechanism of WIP hydrolysis by Alcalase. Structural characteristics showed that the UV fluorescence intensity and endogenous fluorescence intensity of the hydrolysates were significantly higher than those of the control. FTIR results suggested that the secondary structure gradually shifted from an ordered to a disordered structure. Enzymatic hydrolysis containing much smaller molecule peptides than WPI was observed by molecular weight distribution. In vitro, an antioxidant test indicated that Alcalase protease hydrolysis at 120 min showed more potent antioxidant activity than hydrolysates at other hydrolysis times. In addition, four new antioxidant peptides were identified by LC-MS/MS. Molecular docking indicated that these peptides could interact with ABTS through interactions such as hydrogen bonding and hydrophobic interactions. Thus, WPI hydrolysates could be used as potential antioxidants in the food and pharmaceutical industries.

## 1. Introduction

Excess free radicals can lead to oxidative stress, which can cause various chronic diseases such as cardiovascular disease, cancer or diabetes [1,2]. At this point, the presence of antioxidants is significant as a way to prevent oxidative stress and its harmful effects. Antioxidants are subdivided into two main categories, natural and synthetic. Some synthetic antioxidants, although less expensive and more efficient, are somewhat toxic. As a result, there is a growing concern to discover natural antioxidants that are derived from food but have no or few side effects [3]. The public is also increasingly inclined to choose natural antioxidants. A number of antioxidant peptides of plant and animal origin have been reported [4,5,6]. Peptides with high antioxidant capacity were identified from a variety of protein hydrolysates and used as functional food ingredients such as peony [7] and *Torreya grandis* [8]. The common preparation methods of antioxidant peptides include chemical synthesis, and the microbial fermentation and enzymatic hydrolysis of proteins. Enzymatic hydrolysis is the most common method for extracting protein hydrolysates due to its mild and controlled reaction conditions. Depending on the degree of hydrolysis and the molecular weight distribution of the hydrolysate, protein hydrolysates can be classified as mild, moderate and deep hydrolysates. The hydrolysates of deep enzymatic hydrolysis are mainly small peptides and amino acids. The protein enzymatic hydrolysis reaction process is mild and has high product safety. There is no amino acid destruction phenomenon, no environmental pollution and the enzyme hydrolysis process can be controlled. Protein enzymatic hydrolysates with high nutritional value and easy digestion can improve processing functions and significant activity. Zhang et al. reported the use of Neutrase, Alcalase, Flavourzyme and Protamex to obtain salmon skin protein hydrolysates [9]. In recent years, studies on plant protein hydrolysates have mainly focused on antioxidant activity and identification [10,11], with less research on the relationship between their structure and antioxidant activity.

Walnuts (*Juglans regia* L.), also known as pecans and Qiang peaches, belong to the pecan family. The walnut kernel is rich in 18% to 24% protein. In recent years, China’s walnut industry has developed rapidly, ranking first in the world in terms of planted area and production, becoming a veritable powerhouse of walnut production. However, walnuts are often used in the market or for oil extraction and are not fully utilized. Walnut protein contains a high proportion of hydrophobic amino acids, which is closely related to the antioxidant activity of peptides [12]. Therefore, it is necessary to study the structure-activity relationship of the walnut antioxidant peptide to broaden the application of walnut protein.

Our main research process was as follows: (1) A kinetic model for the enzymatic hydrolysis of walnut protein isolates (WPI) was devised and developed. (2) The structure, amino acid composition, molecular weight distribution and antioxidant capacity of walnut protein isolate hydrolysate (WPH) were evaluated. (3) The peptides were identified and molecularly docked to investigate the conformational relationships of the antioxidant peptides. This study provides a theoretical basis for the accurate study of antioxidant peptides and the application of antioxidant peptides from WPI in the food industry and is of great significance for the development of related products.

## 2. Materials and Methods

### 2.1. Materials

Dried walnut was purchased from Shihezi Farmers’ Market (Xinjiang, China). Neutrase (100 U/mg), Alcalase (200 U/mg), Flavourzyme (30 U/mg), Pepsin (500 U/mg) and Protamex (U/mg) with food grade were purchased from Solarbio Technology Co., Ltd. (Beijing, China). All other reagents and chemicals used in this study were of analytical grade and were purchased from Sinopharm Chemical Reagent Co., Ltd. (Shanghai, China).

### 2.2. Extraction of WPI

WPI was extracted by Alcalase solution and acid precipitation. Briefly, shelled and peeled walnuts were crushed and defatted three times by n-hexane, defatted walnuts was dried in the fume hood and screened into defatted walnut powder. Defatted walnut powder was mixed with deionized water at a ratio of 1:10 (*w/v*). pH was adjusted to 9.0 and stirred at room temperature for 1 h. The solution was centrifuged at 8000 r/min at 4 °C for 20 min and the supernatant collected. pH was adjusted to 4.5 and stirred at room temperature for 1 h. The solution was centrifuged again at 8000 r/min at 4 °C for 20 min. The supernatant was collected and lyophilized, and stored at −20 °C for later use.

### 2.3. Preparation of Walnut Protein Hydrolysates (WPH)

WPI was dispersed in deionized water at a substrate concentration of 4% (*w/v*), and the enzyme–protein ratio of 2% (*w/w*). The solution was adjusted to the optimum pH and temperature for the protease. The optimal enzymatic conditions for different enzymes are shown in Table 1. At the end of the enzymatic hydrolysis, the enzyme was inactivated in a boiling water bath for 10 min, then cooled in an ice water bath and centrifuged at 8000 r/min for 20 min. The supernatant was freeze-dried to obtain the walnut protein hydrolysates and stored at −20 °C for future use.

### 2.4. Determination of Degree of Hydrolysis (DH)

pH-stat method [13] was used to determine DH.
(1)$$DH\left(\%\right)=\frac{c\times V}{m\times \alpha \times h_{tot}}\times 100$$
where *V* is the consumption of NaOH solution (mL); *c* is the concentration of NaOH solution (mol/L); *α* is the dissociation degree of α-NH_2_, 1/*α* is the calibration factor, for alkaline protease, 1/*α* is 1.01; *m* is walnut protein quality (g); *h_tot_* is the total number of peptide bonds of walnut protein (mmol/g), the walnut protein was 7.35 mmol/g.

### 2.5. Hydrolysis Kinetic

#### 2.5.1. Enzymatic Hydrolysis

WPI was hydrolyzed at 55 °C with pH = 9.0 by Alcalase, and the degree of hydrolysis with different time under substrate concentrations (S_0_) and enzyme concentrations (E_0_) was studied. WPI was hydrolyzed by Alcalase (0.8 mg/mL) with substrate concentrations of 20, 40, 60 and 80 mg/mL, respectively. In hydrolyzed substrate concentration (40 mg/mL), enzyme concentration was 0.6, 0.8, 1.0 and 1.2 mg/mL, respectively, to study the effect of enzyme concentration on DH.

#### 2.5.2. Hydrolysis Model

The basic process of enzymatic hydrolysis of proteins can be summarized as follows: the formation of the adsorption complex (*ES*) comes from the enzyme (*E*) and the substrate, (*S*) which also known as the Miesian complex; as the protein amide bond breaks and nucleophilic reactions occur, the product (*P*) is released and the enzyme (*E*) regenerates as hydroxyl groups launch an electrophilic attack on the covalently acylase intermediate *ES*.

Alcalase protease is an endonuclease, and the hydrolysis reaction of protein can be expressed as:(2)$$E+S\begin{array}{c} k_{1}\\ ⟺\\ k_{-1} \end{array}ES\begin{array}{c} k_{2}\\ \to \end{array}E+P$$
where *k*_1_ and *k*_−1_ are the rate constants of the reverse reaction, and *k*_2_ is the rate constant. The reaction rate *r* of the enzymatic hydrolysis system depends on the rate of irreversible reaction.
(3)$$r=S_{0}\frac{d\left(DH\right)}{dt}=k_{2}\left[ES\right]$$
where, *S*_0_ is the initial concentration of the substrate, *t* is the reaction time, and *DH* is the degree of hydrolysis.
(4)$$\;r_{ES}=0=k_{1}\left[E\right]\left[S\right]-k_{-1}\left[ES\right]-k_{2}\left[ES\right]$$
(5)$$\left[ES\right]=\frac{\left[E\right]\left[S\right]}{K_{M}}$$
(6)$$K_{M}=\frac{k_{-1}+k_{2}}{k_{1}}$$

The enzyme inactivation mechanism is expressed as:(7)$$E+ES\overset{k3}{\to }Ea+Ei+P$$
where, *E_a_* is the active protease and *E_i_* is the inactive protease.

Then the kinetic equation of the process as:(8)$$-\frac{de}{dt}=k_{3}\left[E\right]\left[ES\right]$$
where e is the total amount of enzyme.

Combination of (3) with (7) provides
(9)$$-S_{0}\frac{d\left(DH\right)}{de}=\frac{k_{2}}{k_{3}\left[E\right]}$$

Because of [*E*_0_] << [*S*_0_], there are [*S*] ≈ [*S*_0_], [P] ≈ [*P*_0_].

When the enzyme is inhibited, it can be expressed as:(10)$$S+ES\begin{array}{c} k_{4}\\ ⟺\\ k_{-4} \end{array}SES$$
(11)$$E+P\begin{array}{c} k_{5}\\ ⟺\\ k_{-5} \end{array}EP$$
(12)$$K_{S}=\frac{k_{-4}}{k_{4}}\;\;\;\;\;K_{P}=\frac{k_{-5}}{k_{5}}$$

The enzyme inactivation rate of the enzymatic hydrolysis system can be expressed as:(13)$$r_{E}=\frac{d\left[Et\right]}{dt}=k_{2}\left[E\right]\left[ES\right]$$
(14)$$r{\;}_{SES}=0=k_{4}\left[S\right]\left[ES\right]-k_{-4}\left[SES\right]$$

Arrange and substitute (5) into:(15)$$\left[SES\right]=\frac{\left[S\right]\left[ES\right]}{K_{S}}=\frac{{\left[S\right]}^{2}\left[E\right]}{K_{M}K_{S}}$$
(16)$$\;\left[EP\right]=\frac{\left[E\right]\left[P\right]}{K_{P}}=\frac{K_{M}\left[ES\right]\left[P\right]}{K_{P}\left[S\right]}$$

It can be obtained by following the conservation of mass in the enzymatic hydrolysis process:(17)$$\left[E_{t}\right]=\left[E\right]+\left[ES\right]+\left[SES\right]+\left[EP\right]$$

Substitute (5), (15) and (16) into (17), and obtain after finishing:(18)$$\left[E\right]=\frac{K_{M}K_{S}K_{P}\left[E_{t}\right]}{K_{S}K_{P}\left[S_{0}\right]+K_{P}{\left[S_{0}\right]}^{2}+K_{M}K_{S}P}$$
(19)$$\left[ES\right]=\frac{K_{S}K_{P}\left[E_{t}\right]\left[S_{0}\right]}{K_{S}K_{P}\left[S_{0}\right]+K_{P}{\left[S_{0}\right]}^{2}+K_{M}K_{S}P}$$

Substitute (18) into (9) and the integral is obtained:(20)$$\;\;\;\;\;\;\;\;\;\;\;\;\;\;\left[E_{t}\right]=[S_{0}]\cdot \exp\left[\frac{k_{3}K_{M}K_{S}K_{P}\left[S_{0}\right]\left(DH\right)}{k_{2}(K_{S}K_{P}\left[S_{0}\right]+K_{P}{\left[S_{0}\right]}^{2}K_{M}K_{S}P)}\right]\;\;\;\;\;\;\;\;\;$$

Substitute (19) and (20) into (3) to obtain:(21)$$DH=\frac{1}{b}\ln\left(1+ab\cdot t\right)$$
$$a=\frac{k_{2}K_{S}K_{P}\left[E_{0}\right]}{K_{S}K_{P}\left[E_{0}\right]+K_{P}{\left[S_{0}\right]}^{2}+K_{M}K_{S}P}\;\;\;\;\;\;b=\frac{k_{3}K_{M}K_{S}K_{P}\left[S_{0}\right]}{k_{2}\left(K_{S}K_{P}\left[S_{0}\right]+K_{P}{\left[S_{0}\right]}^{2}+K_{M}K_{S}P\right)}$$
where if substrate inhibition and product inhibition in the enzymatic hydrolysis system are so small that *K_S_* and *K_P_* can be ignored, parameter A is a linear regression function of ([*E*_0_]/[*S*_0_]), and parameter B is a constant only related to temperature.

Therefore, the a and b values in the model can be determined by the enzymatic hydrolysis time and the corresponding degree of hydrolysis, thereby obtaining the kinetic model equation.

In order to determine the parameters of the kinetic model, the effects of different enzyme dosages and substrate concentrations on the degree of hydrolysis were studied. The *E*_0_/*S*_0_ of the hydrolysis system was adjusted by changing the initial enzyme concentration (*E*_0_) and initial substrate concentration (*S*_0_) of the system. The IBM SPSS Statistics 25 was used to perform non-linear fitting on the test data, and the corresponding kinetic parameters a and b values of different E_0_/S_0_ were obtained.

### 2.6. Amino Acid Composition

Amino acid composition was analyzed by an automatic amino analyzer Model L-8900 (Hitachi, Japan) according to GB/T 5009.124-2016. Protein in food was hydrolyzed into free amino acids by hydrochloric acid, separated by an ion exchange column and then reacted with ninhydrin solution to produce a color reaction, and the amino acid content was determined by visible light spectrophotometry detector. Briefly, the WPHs were hydrolyzed at 110 °C for 22 h using 6 M HCl. After filtration and centrifugation, the amino acid composition of the supernatant was analyzed by automatic amino acid analyzer.

### 2.7. Structural Characterization of WPHs

#### 2.7.1. Ultraviolet Visible (UV) Absorption Spectroscopy

UV-Vis absorption spectroscopy is used to study the microenvironment in which the relevant side chain groups of a sample are located and thus to deduce the general conformation of the protein molecule. In short, for UV spectrum of the WPHs, 1 mg/mL was prepared as the solution with distilled water and scanned with spectra in the range of 190–320 nm.

#### 2.7.2. Intrinsic Fluorescence Spectroscopy

The WPH solutions (0.2 mg/mL) were prepared with 0.01 M phosphate buffer (pH 7.0). The excitation wavelength was set at 285 nm, and emission spectra were recorded at 300–400 nm. Emission and excitation bandwidths were set at 5 nm.

#### 2.7.3. Fourier-Transform Infrared Spectroscopy

The WPHs were crushed together with KBr and were scanned at 1000–4000 cm^−1^ infrared.

### 2.8. Determination of Antioxidant Activities

#### 2.8.1. DPPH Radical Scavenging Activity

The DPPH radical scavenging activity was carried out using the method of Dong et al. [14] with slight changes. The 0.1 mmol/L DPPH solution was prepared with anhydrous ethanol, and 1 mL of the test sample solution was added to 1 mL DPPH solution, mixed well, and the absorbance was measured at 517 nm after standing for 30 min in the dark at chamber temperature, and 1 mL DPPH was measured at the same time. The absorbance of the solution mixed with 1 mL ethanol and the absorbance of the test sample solution mixed with 1 mL anhydrous ethyl alcohol was calculated.
(22)$$DPPH•\mathrm{scavenging}\;\mathrm{activity}\;\left(\%\right)=\left(1-\frac{A_{1}-A_{2}}{A_{0}}\right)\times 100$$
where *A*_0_ is the absorbance of the reaction with other reagents, *A*_1_ is the absorbance of the sample with other reagents and *A*_2_ is the absorbance of the sample.

#### 2.8.2. ABTS^+^• Scavenging Activity

The ABTS^+^• scavenging activity was assayed following the method of Wang et al. [15] with minor modifications. ABTS^+^• was produced by mixing 7 mmol/L ABTS with 2.45 mmol/L potassium persulfate solution and reacting at room temperature under dark conditions for 12–16 h. Before use, solution obtained above was diluted with deionized water, and its absorbance at 734 nm was 0.70 ± 0.02. The 50 μL samples were mixed with 150 μL ABTS solution above, and the absorbance was measured at 734 nm.
(23)$$ABTS^{+}•\mathrm{scavenging}\;\mathrm{activity}\;\left(\%\right)=\left(1-\frac{A_{1}-A_{2}}{A_{0}}\right)\times 100$$
where *A*_0_ is the absorbance of the reaction with other reagents, *A*_1_ is the absorbance of the sample with other reagents and *A*_2_ is the absorbance of the sample.

#### 2.8.3. Hydroxyl Radical Scavenging Activity

The hydroxyl radical (OH•) scavenging activity was determined by Song et al. [16] with some modification. Briefly, the reaction mixture contained 1 mL WPH, 1 mL ferrous sulfate (3 mM) and 1 mL salicylic acid–ethanol (3 mM). The reaction began with the addition of 1 mL hydrogen peroxide (6 mM). After incubation at 37 °C for 30 min, the absorbance of the solution was measured at 510 nm. The OH• scavenging activity was expressed as the scavenging rate and was calculated using the following formula:(24)$$OH•\mathrm{scavenging}\;\mathrm{activity}\;\left(\%\right)=\left(1-\frac{A_{1}-A_{2}}{A_{0}}\right)\times 100$$
where *A*_0_ is the absorbance of the reaction with other reagents, *A*_1_ is the absorbance of the sample with other reagents and *A*_2_ is the absorbance of the sample.

#### 2.8.4. Reducing Power

The reducing power was measured using the method of An et al. [17] with minimal adaptation. In total, 1 mL of the sample was added to 1 mL of 1% (*w/v*) potassium ferricyanide with 1 mL of 0.2 M PBS, and the reaction was terminated by adding 1 mL of 10% TCA solution at 50 °C for 20 min. Centrifuged at 4000 r/min for 10 min, 2.5 mL supernatant was evenly mixed with 1.2 mL 0.1% ferric chloride and reacted at room temperature for 10 min. Absorbance value was measured at 700 nm.

### 2.9. Determination of Molecular Weight Distribution

The molecular weight (MW) distribution of WPHs was analyzed by high-performance liquid chromatography system (HPLC; Waters e2686, Milford, MA, USA) with a TSK gel G2000 SW_XL_ column (Tosoh, Tokyo, Japan) as described by Cui et al. [18]. The equipment was purchased from Water Technology (Shanghai) Co., Ltd. (Shanghai, China) The column was eluted with acetonitrile/water/trifluoroacetic acid = 45/55/0.1 (*v/v/v*) at a flow rate of 0.5 mL/min and the eluent was monitored at 214 nm [19]. The MW of peptides was calculated based on the calibration curve constructed with cytochrome C (MW, 12,500 Da), aprotinin (MW, 6500 Da), bacitracin (MW, 1450 Da), Gly-Gly-Tyr-Arg (MW, 451.2 Da), H-Gly-Gly-Gly-OH and (MW, 189.17 Da).

### 2.10. Separation and Purification of WPH

The WPH was separated and purified by gel filtration chromatography (hydrolysis conditions: Alcalase protease, mass fraction 4%, substrate mass fraction 3%, temperature 55 °C, pH 9.0, hydrolysis for 2 h). DEAE cellulose −52 filler was selected, and the treated filler was loaded into the chromatography column. After the gel column was pretreated, the sample was added slowly with a mass concentration of 50 mg/mL and injection volume of 10 mL. The eluent was deionized water, and the elution flow was controlled at 0.7 mL/min. One tube was collected every 10 min, and the UV detection wavelength was 280 nm. The eluent at each peak was collected, lyophilized and stored at −20 °C until further use.

### 2.11. Identification of Peptides of WPH by LC-MS/MS

The peptides from the separation and purification of WPH were identified by LC-MS/MS. Sequencing was performed using Q Exactive™ Hybrid Quadrupole-Orbitrap™ Mass Spectrometer (Thermo Fisher Scientific, USA). A pre-column (300 μm i.d. × 5 mm, packed with Acclaim PepMap RPLC C18, 5 μm, 100 Å), an analytical column (150 μm i.d. × 150 mm, packed with Acclaim PepMap RPLC C18, 1.9 μm, 100 Å) with a flow rate of 600 nL/min with a gradient of 60 min (0 min 6% B; 5 min 9% B; 20 min 14% B; 50 min 30% B; 58 min 40% B; 60 min 95% B; mobile phase A containing 0.1% formic acid and 80% ACN. mobile phase B containing 0.1% formic acid and 97% ACN. mobile phase A containing 0.1% formic acid and 80% ACN. MS resolution: 70,000 at 400 *m*/*z*, MS precursor *m*/*z* range: 300.0–1800.0. Product ion scan range: start from *m*/*z* 100, Activation Type: CID, Min. Signal Required: 1500.0, Isolation Width: 3.00, Normalized Coll. Energy: 40.0.

### 2.12. Bioinformatics Determination of Physicochemical Properties of Antioxidant Peptides

The relevant properties of peptides can be analyzed by online tools such as PeptideRanker (http://distilldeep.ucd.ie/PeptideRanker/; accessed on 15 March 2022) which ranks peptides according to their predicted probability of being biologically active. ToxinPred (http://crdd.osdd.net/raghava/toxinpred/; accessed on 15 March 2022) allows the toxicity prediction of peptides. Expasy (https://web.expasy.org; accessed on 15 March 2022) calculates the theoretical pI and molecular weight Mw of peptides, various physical and chemical parameters of a given protein stored in Swiss-Prot or TrEMBL or a protein sequence entered by the user, calculating parameters such as molecular weight, theoretical pI, amino acid composition, atomic composition, extinction coefficient estimated half-life, instability index, lipid index and hydrophilic mean. PepDraw (http://pepdraw.com/; accessed on 15 March 2022) draws the primary structure of the peptide and calculates the theoretical properties (isoelectric point, net charge, hydrophobicity and extinction coefficient).

### 2.13. Molecular Docking

The 3D structure of ABTS (CID: 5360881) was obtained from the PubChem database (https://pubchem.ncbi.nlm.nih.gov/; accessed on 10 April 2022). The 3D structures of the novel peptides identified by mass spectrometry were obtained through homology modelling by Modeller. The ABTS was docked using Autodock software and its structure was optimized using a genetic algorithm based on the principle of minimal docking energy. The steric and planar interactions of the receptor peptide with the ligand were analyzed using PyMOL and Liglot software.

### 2.14. Statistical Analysis

All experiments were performed using at least three replicate experiments. All data were reported as mean standard deviation. Data analyses were performed using IBM SPSS Statistics 25 (SPSS, Chicago, IL, USA). Statistical significance was determined using Duncan’s multiple range tests with *p* < 0.05 used as the threshold for statistical significance.

## 3. Results and Discussion

### 3.1. Screening of Enzymes

DH is a measure of the rate of protein hydrolysis and captures the amount of peptide bond breakage in the protein [20]. The choice of the enzyme is key in the proteolytic process, which determines the location and number of hydrolyze peptide bonds and affects the reaction rate, yield and biological activity of the final product. Different enzymes hydrolyze different parts of the protein peptide chain depending on the specificity of the acting group and thus produce different enzymatic products. It is generally accepted that the specificity of the enzyme to the substrate significantly influences the molecular weight size and the hydrophilicity or hydrophobicity of the enzymatic product. The optimum enzymatic temperature and pH for the five enzymes are shown in Table 1. The best proteases are screened by DH and DPPH radical scavenging rates to obtain biologically active enzymatic products. As shown in Figure 1, there were significant differences in DH values between the groups after hydrolysis by different proteases (*p* < 0.05). Obviously, the Alcalase protease had the highest DH (12.03%) for the same hydrolysis time. In addition, the antioxidant properties of the five enzymatic products were measured and the results showed that the Alcalase protease had the best DPPH radical scavenging activity (86.28%). The reason for this result may be that Alcalase proteases have a wide range of sites of action and work well with both hydrophilic and hydrophobic amino acids. The main sites of action of trypsin are the carboxyl and amino groups of basic amino acids such as lysine and arginine. Taking everything into consideration, Alcalase protease was finally chosen for the hydrolysis of WPI.

### 3.2. The Enzymolysis Kinetic Model of WPI

Enzymolysis kinetics mainly studies the enzymolysis mechanism and the intrinsic rules affecting the enzymolysis rate. The establishment of the enzymolysis kinetics model is of great significance to the control of enzymolysis process conditions. Figure 2 shows the variation in the hydrolysis degree of WPI within the 180 min hydrolysis process. It is clear that the degree of hydrolysis of WPI increased with increasing protease concentration and decreased with increasing protein concentration. Furthermore, the degree of hydrolysis of WPI increased rapidly during the first hour and increased slowly during the next two hours. In our study, the trend in the degree of enzymatic hydrolysis of WPI is divided into three phases with increasing hydrolysis time: a rapid increase phase, a slow growth phase and an equilibrium phase. At the beginning of the hydrolysis phase, the enzymatic system has a low nitrogen extraction rate, mainly from the water-soluble proteins of the protein feedstock itself. This is followed by rapid degradation of the protein to peptides and amino acids by the action of proteases. However, as the protein hydrolysis reaction goes on, the decreasing concentration of biological proteins in the hydrolysis system means that the concentration of protein substrates available to the protease also decreases, resulting in a slowdown in the rate of the hydrolysis of WPI. The trend of the enzymatic reaction of WPI in this study was consistent with the hydrolysis trends of peony seed protein isolate [7] and *Torreya grandis* meal protein [8].

The values of the kinetic parameters a and b were calculated by Equation (21) and the results are shown in Table 2. The experimental data show that the value of a is not a constant but has a good linear relationship with E_0_/S_0_, followed by a = 24.85 (E_0_/S_0_) + 0.31 (R^2^ = 0.9929). Parameter b fluctuates only within a small range, so the average value of 0.269 is taken as b. Substituting the values of parameters a and b into Equation (21), the equation of DH was obtained as: DH = 3.72 ln [1 + (6.68 E_0_/S_0_ + 0.08) t].

To demonstrate the reliability of the kinetic equations for enzymatic digestion, three sets of validation experiments with different enzyme and substrate concentrations were designed, as shown in Figure 3. The predicted and experimental values of hydrolysis correlated well, which indicated that the hydrolysis kinetic model for WPI was reliable. Therefore, the degree of hydrolysis of WPI can be controlled according to the kinetic model so as to achieve the purpose of controlling the enzymatic hydrolysis process to obtain the target hydrolysate, which has a good practical application value, and has a certain guiding role in the industrialization of walnut peptide.

### 3.3. Amino Acid Composition Analysis

The amino acid composition of WPI and WPH at different hydrolysis times is shown in Table 3. The main amino acids in the walnut samples were glutamic acid and arginine, accounting for about 21–23% and 14–15% of the total amino acids, respectively. The amino acid pattern of the WPH was slightly different compared to the control (WPI). Studies have reported that the negatively charged acidic amino acids, glutamic acid and aspartic acid, have free radical scavenging capacity owing to the presence of excess electrons [12]. WPH contains aspartic acid and glutamic acid, which make up about 32% of the total amino acids. Therefore, WPH could scavenge free radicals due to the abundance of glutamate and arginine. This was also confirmed by the subsequent determination of free radical scavenging activities (DPPH, ABTS and OH).

Moreover, hydrophobic amino acids of WPHs such as leucine (6.64~7.15%), proline (5.09~5.91%), phenylalanine (4.43~4.57%), valine (4.70~5.07%) and alanine (4.32~4.63%) were high at different hydrolysis times, accounting for about 22% of the total amino acids. Meanwhile, the contents of the hydrophobic amino acids of WPHs increased with the increase in hydrolysis time and were highest with a hydrolysis time of 120 min. Research has revealed that the high proportion of hydrophobic amino acids may promote the proximity of the peptides to lipid radicals [21], thereby enhancing antioxidant activity. The high percentage of hydrophobic amino acids in peptides with high antioxidant activity may be a key factor in the ability of peptides to scavenge free radicals [12]. The hydrophobic amino acid content was positively correlated with the antioxidant activity of polypeptides, so walnut peptides have potential as food-borne bioactive peptides. Additionally, it can be seen that the content of sulfur-containing amino acids increases in WPH compared to WPI; a similar result has been reported by Ghribi et al. [22]. Studies have shown that sulfur-containing amino acids as antioxidants play an important role in the body [23]. In this study, it was obvious that the sulfur-containing amino acids in WPH were higher than those in WPI, and the content was the highest when the hydrolysis time was 120 min.

Wade et al. [24] reported that histidine contributes to the body’s physiological antioxidant capacity. We further discovered the fact that the histidine (His) content increases with the increase in hydrolysis time, which indicates that WPH has good antioxidant activity. According to Song et al., the hydrophobic amino acids histidine (His), valine (Val) and phenylalanine (Phe), the hydrophilic and basic amino acids histidine (His), proline (Pro) and lysine (Lys) and the aromatic amino acids phenylalanine (Phe) and Y contained in the antioxidant peptides were found to be responsible for their high antioxidant activity [16]. Interestingly, the L (hydrophobic amino acid), F (hydrophobic amino acids/aromatic amino acid) and P (basic amino acid) accounted for a large proportion among the 10 peptides identified in our study, which further demonstrated the high antioxidant activity of WPH. In addition, walnut peptides all contain a certain amount of tyrosine and methionine, and these amino acids also have obvious antioxidant activity in the short peptide sequence [21].

In summary, WPH with high hydrophobic amino acid content has better antioxidant activity in this study. In addition, the increase in histidine and sulfur-containing amino acids promoted the antioxidation activity.

### 3.4. Structural Characteristics by Spectroscopy

#### 3.4.1. UV-Visible Absorption Spectroscopy

Figure 4A shows the UV-Vis spectra of the WPH and characterizes them in relation to the time of hydrolysis. These walnut protein hydrolysates show a resemblance to the shape of the UV-Vis spectrum in the wavelength range of 190~320 nm. The UV spectra of the hydrolysates all exhibited a maximum absorption peak at 192 nm and an absorption peak at 203~210 nm, responding to the leap in electron-containing π electrons such as C=O, C=C, C≡O, etc. All the enzymatic hydrolysates showed a significant color enhancement effect compared to the WPI due to the exposure of chromophores and changes in protein conformation resulting from the enzymatic treatment. Furthermore, the increase in UV absorption values as the hydrolysis time was extended from 30 min to 120 min, although it then declined with further increases in hydrolysis time, is likely due to the release of amino acid residues from the protein interior into the water and the increase in the number of chromophores as the hydrolysis process was prolonged. Similarly, Zhang et al. reported that enzymatic digestion significantly enhanced the strength of hydrolysis products by exposing aromatic chromophores and hydrophobic groups [9].

#### 3.4.2. Fluorescence Spectra Analysis

Fluorescence spectrometry is a valid method for protein conformation analysis.

The structure changes in WPI and WPH were studied by endogenous fluorescence spectroscopy. Among protein molecules, amino acids that emit fluorescence are tryptophan (Try), tyrosine (Tyr) and phenylalanine (Phe). As shown in Figure 4B, the fluorescence intensity peak of WPI was observed at 336 nm, while the fluorescence intensity peak of WPI was observed at 423 nm, indicating a phenomenon of red shift. The fluorescence intensity of WPH was higher than that of the control (WPI), and increased with the increase in hydrolysis time until 150 min. A similar behavior was observed in oat proteins hydrolysis with Alcalase [21]. Protein hydrolysis leads to the exposure of tryptophan residues and increased fluorescence intensity [25]. Nevertheless, the fluorescence intensity of WPH decreased at a hydrolysis time of 150 min, probably owing to the reburial of some of the hydrophobic amino acids on the peptide surface and inside the peptide.

#### 3.4.3. FTIR Spectra Analysis

The FTIR spectra of the enzymatically-treated WPH samples show some similarity in terms of peak shapes (Figure 4C), but they differ in terms of absorption intensities. We can see clearly that the intensity of the absorption peak of WPH increased with different enzymatic hydrolysis times. The peak at around 1662 cm^−1^ in the sample is consistent with the region of the amide I band, which is the consequence of the C=O stretching vibration of the peptide bond [26].

The protein secondary structure is generally evaluated by the amideⅠband. After simulating the peak area of the amide I band, the proportions of the secondary structure of WPI hydrolysates are listed in Table 4. Many studies have shown that the antioxidant’s activity was related to the change in its secondary structure. Compared with the WPI, with the increase in enzymatic hydrolysis time, the ordered structure such as α-helical and β-sheet gradually changed to β-turn and random coil, resulting in the exposure of the active site in the peptide chain, thus improving the antioxidant activity of the hydrolysate. The secondary structure of the skin protein hydrolysate had a tendency to convert α-helical and β-sheet into β-turn, disrupting their ordered structure and making their molecular structure flexible and loose, leading to a change in protein conformation [9].

### 3.5. Molecular Weight Distribution

According to the size exclusion chromatogram of standards, the relative molecular mass correction curve equation plotted with a log of relative molecular mass (lgMW) vs. retention time (T) was obtained.
(25)lgMW=0.2248 T+6.7937

High R^2^ at 0.99301 suggests a good linear of the formula, and can be employed to calculate the molecular weight distribution of each component in the sample.

Several studies have shown that the antioxidant activity of the hydrolysates is closely correlated with their molecular weight. The molecular weight distribution of WPH under different enzymatic hydrolysis times is shown in Figure 5. As can be seen from Figure 5, The proportion of >Da was the highest (85%), which was significantly reduced after enzymatic hydrolysis in our study. Furthermore, the proportion of low molecular weight (<1000 Da) peptides increased with increasing hydrolysis time obviously. It has been reported that an appropriate low molecular weight can contribute significantly to the antioxidant activity of the peptide [12]. Wang et al. found that the molecular weight distribution of proteolytic products of broken rice with different hydrolysis times was different, and the increase in oligopeptide and free amino acid contents increased with the increase in hydrolysis time [27]. This is consistent with the results of this study. In addition, subsequent experimental data demonstrated that the WPH at a hydrolysis time of 120 min had the highest antioxidant activity.

### 3.6. In Vitro Antioxidant Activities

Four in vitro assays (DPPH radical scavenging, ABTS radical scavenging, hydroxyl radical scavenging and reducing power) were used to assess the antioxidant activity of the WPH at different enzymatic hydrolysis times and the results are shown in Figure 6.

#### 3.6.1. DPPH Radical Scavenging Activity

It can clearly be seen that the antioxidant activity of WPH is higher than WPI at different times from Figure 6. What is more, there is a significant positive correlation between hydrolysis time and the DPPH radical scavenging activity of WPH. Our study is in agreement with the study by OLFAOUSSAIEF, where the antioxidant activity of all dromedary colostrum hydrolysates were significantly more effective than single peak colostrum proteins [28]. Similarly, Zheng et al. [21] demonstrated that the rate of free radical scavenging decreased with increasing hydrolysis, indicating a negative correlation between DH and DPPH free radical scavenging activity in oat protein hydrolysis products. It was reported that the DPPH scavenging activity was significantly positively correlated with the total amount of hydrophobic amino acids [29]. Similarly, the higher DPPH radical scavenging activity of WPH for 120 min was positively correlated with its hydrophobic amino acid content. The result corresponds with amino acid composition analysis. The difference in amino acid composition may be one of the reasons for the difference in the antioxidant capacity of WPHs under different hydrolysis times. It has been reported that *T. grandis* meal protein hydrolysate containing a high proportion of electron-donating amino acids has high antioxidant activity [8]. In our study, the previous amino acid composition data showed that DEAA content increased with the prolongation of hydrolysis time, indicating that antioxidant activity also increased.

#### 3.6.2. ABTS Radical Scavenging Activity

Water-soluble ABTS free radicals easily react with antioxidants by transferring protons, thus playing an antioxidant role [25]. As can be seen from Figure 6, ABTS scavenging activity gradually increased to 83.54% when the hydrolysis time increased to 120 min. The maximum scavenging capacity of WPI on ABTS radical was about 62.29%, which was significantly lower than that under the condition of 120 min hydrolysis. This may be due to the high proportion of small peptides (<1000 Da), which endows WPH with strong ABTS radical scavenging activity. However, the scavenging activity decreased after further hydrolysis to 150 min. According to the research report of OUSSAIEF et al. [28], exposure to aromatic and hydrophobic amino acid residues resulted in increased antioxidant activity of the colostrum hydrolysate of dromedary (Camelus Dromedarius). This is consistent with the results of previous amino acid composition analyses of WPH.

#### 3.6.3. Hydroxyl Radical Scavenging Activity

Hydrogen peroxide and superoxide anions can be converted to hydroxyl radicals in the presence of metal ions in the body, and can also be produced by several other free radical systems leading to oxidative stress. Therefore, scavenging hydroxyl radicals is an effective defense strategy of the human body against various diseases caused or propagated by ROS. Figure 6 reveals the scavenging effect of WPH on hydroxyl radicals. The scavenging effect of WPH on hydroxyl radicals increased with the hydrolysis time. When the hydrolysis time was 120 min, the scavenging ability of WPH to hydroxyl radicals reached 82.55%. Likewise, some studies have shown that the antioxidant activity is continuously formed by proteolysis [25]. The amino acid composition data in our study showed that the Met content was the highest when WPH was hydrolyzed for 120 min. Some studies have shown that Met has special structural characteristics and is easily oxidized to sulfoxide, thus it has an antioxidant effect [30]. This is consistent with the result of the hydroxyl radical scavenging activity of WPH.

#### 3.6.4. Reducing Power

It can be seen that the WPH had higher antioxidant activity than the WPI. Moreover, the reducing power increased with the increase in enzymatic hydrolysis time and reached the maximum value at 120 min (0.37 ± 0.03). This might be the result of the increased availability of hydrogen ions using smaller peptides [31]. In contrast to WPI, WPH have a higher ability to terminate free radical chain reactions [32]. Current research results indicate that the smaller the molecular weight and the better the hydrophobicity of the protein, the stronger the ability to scavenge free radicals. This may be due to the fragmentation of the polypeptide chain of the protein under the action of enzymatic hydrolysis, resulting in changes in molecular size and structure. The active region or polypeptide is released, and the antioxidant activity of the protein increases correspondingly. Accordingly, Tang et al. [33] observed that Alcalase enzymatically hydrolyzed the hemp protein for 2 h with the highest reducing power. It has been reported that the reducing power of wheat germ protein hydrolysates (WGPH) may contribute significantly to antioxidant effects [34].

### 3.7. Separation and Purification of WPH

The WPH at 120 min was divided into two fractions (F1 and F2) by DEAE cellulose-52 filtration (Figure 7A). The scavenging ability of each fraction against DPPH radicals was determined; as shown in Figure 7B, F1 had the highest antioxidant activity compared to F2. A number of studies have demonstrated that the low molecular weight fraction of the purified hydrolysate has a significantly greater antioxidant capacity than the high molecular weight fraction [35]. Subsequently, F1 was subjected to LC-MS/MS for peptide sequence identification.

### 3.8. Identification of Antioxidant Peptides by LC-MS/MS

Figure 8 shows a total ion chromatogram. Figure 9 shows the secondary mass spectra of some of the peptides. In the spectra, the vertical and horizontal coordinates are the ion abundance and ion mass ratio (*m/z*), respectively. As most antioxidant peptides contain no more than 10 amino acids, to save time and cost, antioxidant peptides with ≤10 amino acids were considered. Each peptide was scored using PeptideRanker (http://distilldeep.ucd.ie/PeptideRanker/ accessed on 19 September 2022) to evaluate the probability of biological activity of the peptide. In this study, peptides with a PeptideRanker score higher than 0.5 were selected for further study. Ten peptides with a PeptideRanker score greater than 0.5 and fewer than 10 amino acid residues were used.

The specific information of the 10 selected antioxidant peptides is shown in Table 5. The table shows us that the molecular weight of the peptide was between 604.312 Da and 920.420 Da. The antioxidant activity is related to the composition, quantity and sequence of some amino acids in the peptide fragment, among which histidine residue, proline residue, cysteine residue, leucine residue and the hydrophobicity of the peptide all have a great influence on the antioxidant activity. Peptides containing hydrophobic amino acid residues at the N-terminal have better antioxidant capacity. It has been reported that the majority of antioxidant peptides (<3 kDa) contain hydrophobic amino acids such as Gly, Tyr, Leu, Met, Pro, Trp, Val and Phe residues at the N- or C-terminus of their sequences [35]. In our study, the C- or N-terminus of the ten antioxidant peptides all contained hydrophobic amino acids except HADMVFY. Part of the peptides is involved in hydrogen atom transfer and single electron transfer reactions through the imidazole group of histidine, contributing to the antioxidant effect. Among the 10 identified peptides in our study, we observed the presence of histidine in peptide HADMVFY, NHCQYYL and NLFHKRP.

In addition, the chemical structures of the ten peptides are shown in Figure 10. According to Habinshuti et al. [36], a bulky hydrophobic amino acid (W, Y, F, M, L or I) with low electron or space/hydrogen bonding properties at the C-terminal third position of the peptide may be responsible for the antioxidant activity. The presence of aromatic amino acids such as phenylalanine allows the direct transfer of electrons to reactive oxygen species (ROS), thus playing an important role in the antioxidant activity of the peptide [37]. In the present study, the peptides isolated contained aromatic amino acids such as phenylalanine, tryptophan or tyrosine, with the exception of QAGQLLPL and SNAPQL. Proline and threonine have been reported to act as antioxidants due to their unique structures, namely the pyrrolidine ring (proline) and the indole ring (threonine) [38]. Among the 10 peptides identified from WPH, we found the presence of proline or threonine except in HADMVFY and NHCQYYL.

Through the toxicity prediction of professional websites, these peptides are not toxic and have high safety. However, the prediction results used in the experiment were obtained by using the database. Further studies on the toxicity of synthetic peptides in humans require either cellular or in vivo tests. Furthermore, the amphiphilic character of the peptide also appears to promote free radical scavenging activity by improving the solubility of the peptide while fostering interactions with free radicals and proton exchange [39]. This conclusion is consistent with the previous results of WPH antioxidant determination. Moreover, all the peptides in Table 6 had positive hydrogen bond numbers and the hydrophobicity calculated by the database was high, which played a positive role in the antioxidant activity of WPH.

### 3.9. Molecular Docking Analysis

The minimum binding energy for peptide and ABTS docking is shown in Table 7.

As shown in Figure 11, the optimal binding energy configurations for HADMVFY, NHCQYYL, NLFHKRP and PSYQPTP and with ABTS were obtained based on the minimum binding energy after docking and the minimum binding energies were 4.263, 4.379, 4.401 and 4.423 kcal/mol, respectively. All peptides can form hydrogen bonds and hydrophobic interactions with residues in ABTS. In HADMVFY, a hydrogen bond of 3.07 Å was formed between the Tyr7 residue and ABTS, and a hydrogen bond of 2.89 Å was formed between the Asp3 residue and ABTS, and hydrophobic interactions occurred with Phe6 and Ala2 residues. The Tyr3 residue of PSYQPTP formed a hydrogen bond of 2.98 Å with ABTS, and Prol formed a 2.91 Å hydrogen bond with ABTS. Residues Pro5, Thr6 and Pro7 formed three hydrophobic interactions with ABTS. The Tyr6 residue of NHCQYYL formed 2.98 Å hydrogen bond with ABTS. The Cys3 residue formed 3.02 Å hydrogen bond with ABTS. ABTS had hydrophobic interactions with Leu7, Gln4 and Tyr5 residues of NHCQYYL. Lys5 residues of NLFHKRP interact with ABTS via hydrogen bonds with a bond length of 3.10 Å. In addition, Arg6, Phe3, Leu2 and His4 residues formed four hydrophobic interactions with ABTS.

In addition, residues Arg-6, Phe-3, Leu-2 and His-4 each interacted hydrophobically with ABTS. Similarly, some amino acid residues of watermelon seed antioxidant peptides interact with ABTS through hydrogen bonding and hydrophobic interactions to scavenge free radicals [40]. These results demonstrate that molecular docking can be applied to study the conformational relationships between peptides and small molecules (e.g., ABTS).

## 4. Conclusions

The present study was designed to investigate the structure-activity relationship of walnut antioxidant peptide. This study demonstrates that the different degree of Alcalase hydrolysis can significantly alter the structure and antioxidant activity of WPI. To be specific, the amino acid patterns of the WPH differed slightly for different hydrolysis times, with a slightly higher content of hydrophobic and hydrophilic amino acids at a hydrolysis time of 120 min. The WPHs had a low molecular weight (<1000 Da) structure and good antioxidant activity as well as high UV absorption and fluorescence intensity. However, attention should be focused on the negative impact of excessive hydrolysis (150 min) on the antioxidant properties of the WPH. In addition, the relationship between the structural characteristics of peptides identified from WPH and their antioxidant activity was also studied, namely, peptides containing hydrophobic amino acids at the C-terminal or N-terminal showed high antioxidant activity. Moreover, the molecular docking results showed that some amino acid residues of the four peptides interact with ABTS through hydrogen bonding and hydrophobicity to play an antioxidant role.

The above findings suggest that our results can identify the key active sites and structure-activity relationships of four novel antioxidant peptides, and supply new ideas for the engineering design and mechanism study of antioxidant peptides. In conclusion, WPHs have the potential as natural antioxidants for functional food and health care products.

## Figures and Tables

**Figure 1 molecules-27-08423-f001:**
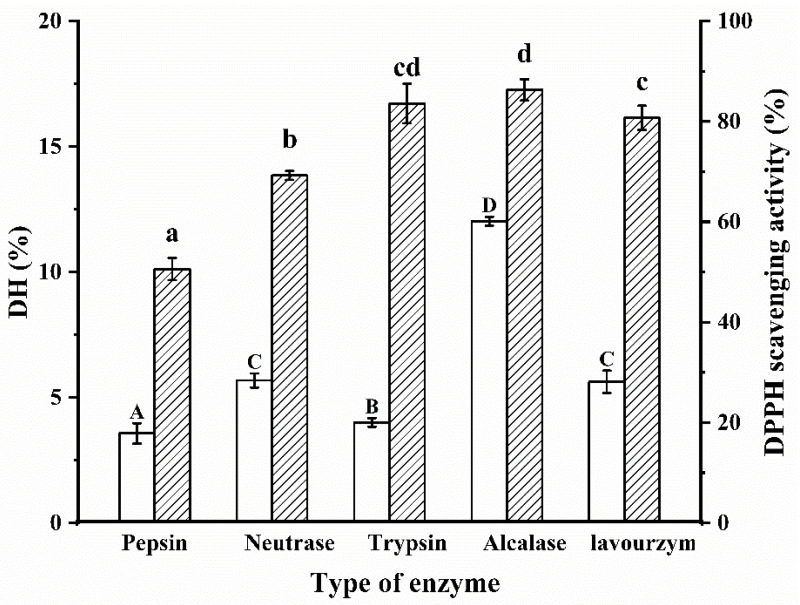
Protease screening. Different letters indicate significant differences in the same indica-tor. Capital letters represent the significant difference of DH of WPI under different proteases, and lowercase letters represent the significant difference of DPPH radical scavenging activity of WPI under different proteases.

**Figure 2 molecules-27-08423-f002:**
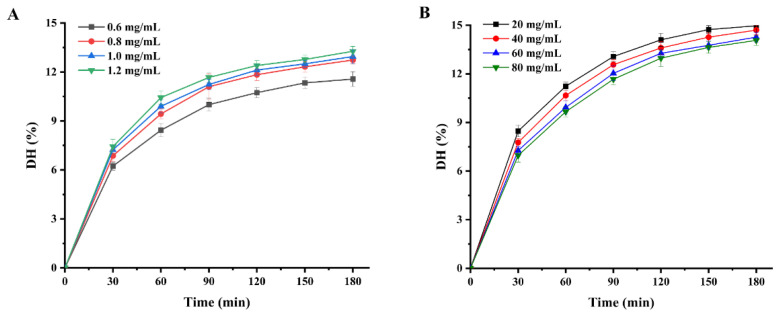
Enzymatic profiles of WPI hydrolysis by Alcalase protease: (**A**) degree of hydrolysis of WPI at different Alcalase protease concentrations; (**B**) DHs of WPI at different substrate concentrations.

**Figure 3 molecules-27-08423-f003:**
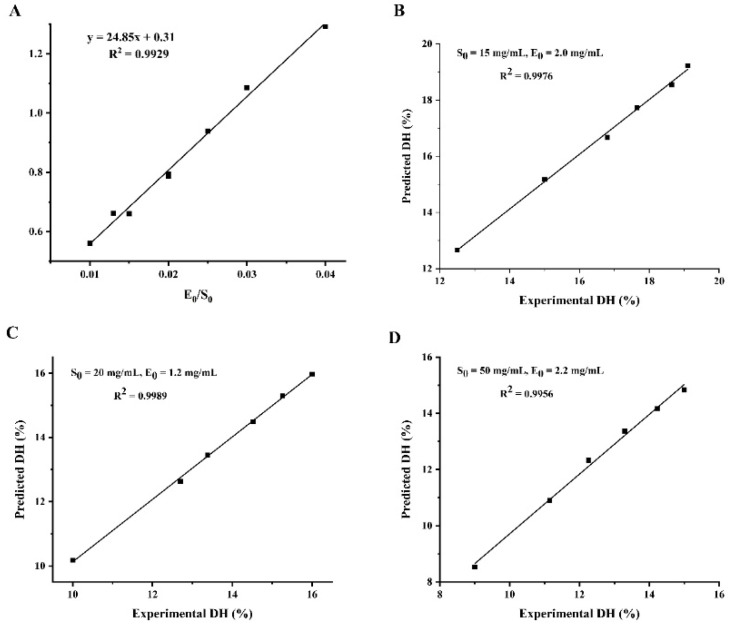
Validation of the WIP hydrolysis kinetic model. (**A**) Linear relationship between parameter a and E_0_/S_0_; (**B**–**D**): correlation between predicted and actual values of hydrolysis under different enzyme and substrate concentrations. (**B**) S_0_ = 15 mg/mL, E_0_ = 2.0 mg/mL; (**C**) S_0_ = 20 mg/mL, E_0_ = 1.2 mg/mL; (**D**) S_0_ = 50 mg/mL, E_0_ = 2.2 mg/mL.

**Figure 4 molecules-27-08423-f004:**
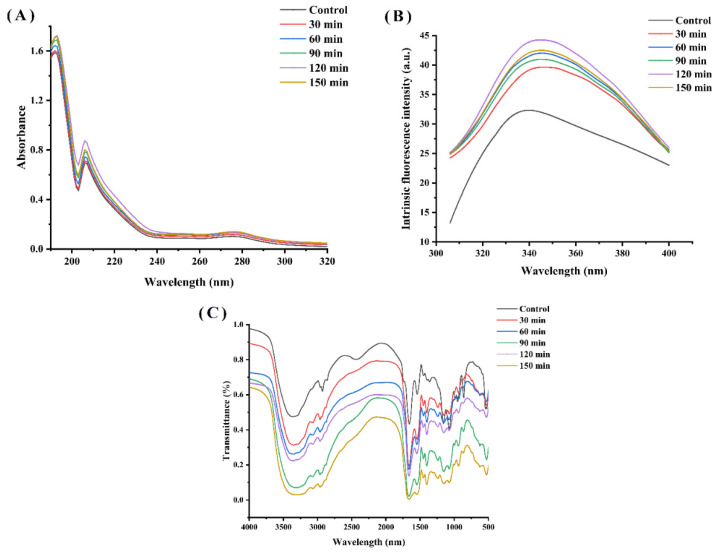
(**A**) UV absorption spectra, (**B**) endogenous fluorescence spectra and (**C**) FTIR spectra of walnut proteins with different enzymatic digestion times.

**Figure 5 molecules-27-08423-f005:**
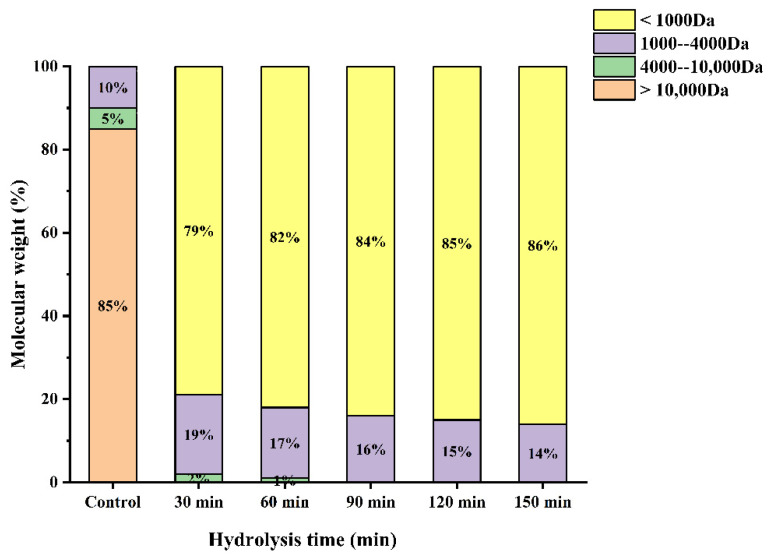
Molecular weight distribution of enzymatic products at different enzymatic times.

**Figure 6 molecules-27-08423-f006:**
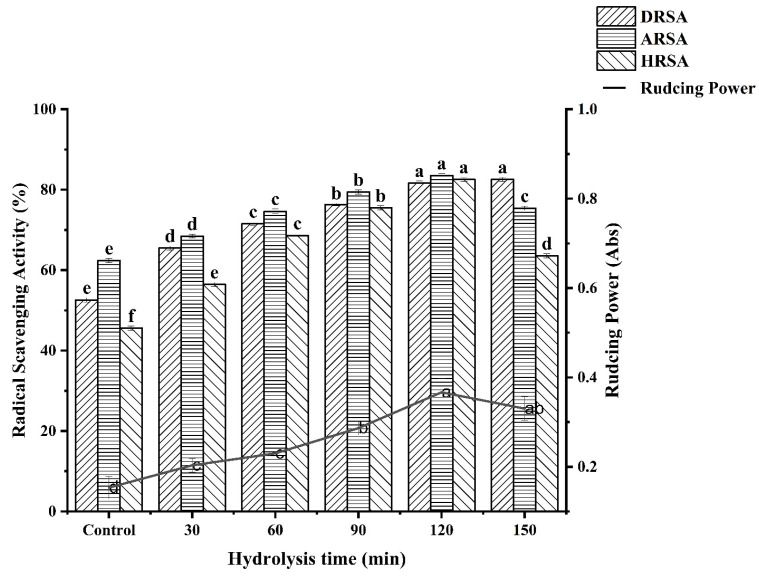
Determination of antioxidant activity at different hydrolysis times. Different letters indicate a significant difference (*p* < 0.05).

**Figure 7 molecules-27-08423-f007:**
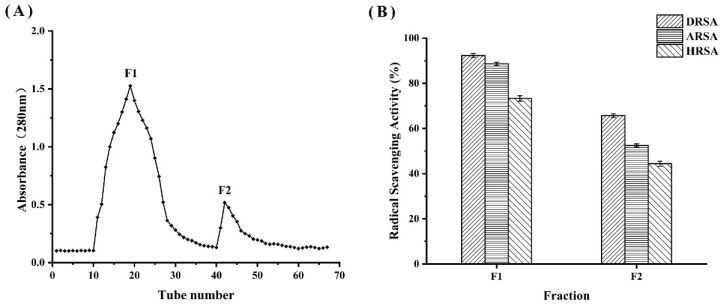
(**A**) Filtration chromatography by DEAE cellulose-52 and (**B**) determination of the antioxidant activity of the F2 fraction.

**Figure 8 molecules-27-08423-f008:**
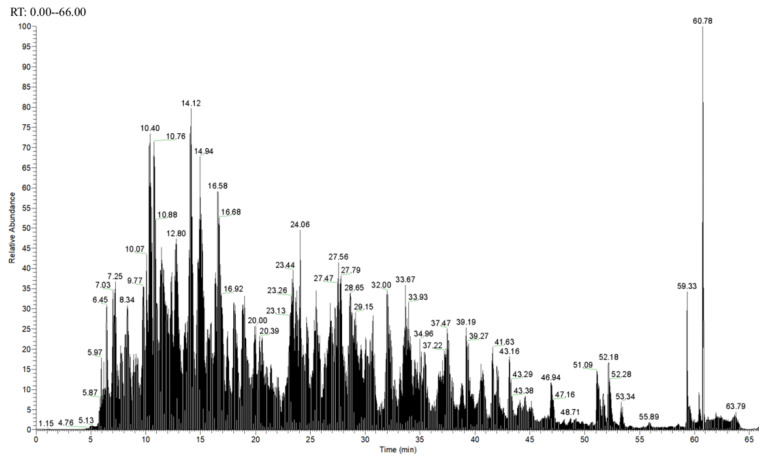
Total ion chromatogram.

**Figure 9 molecules-27-08423-f009:**
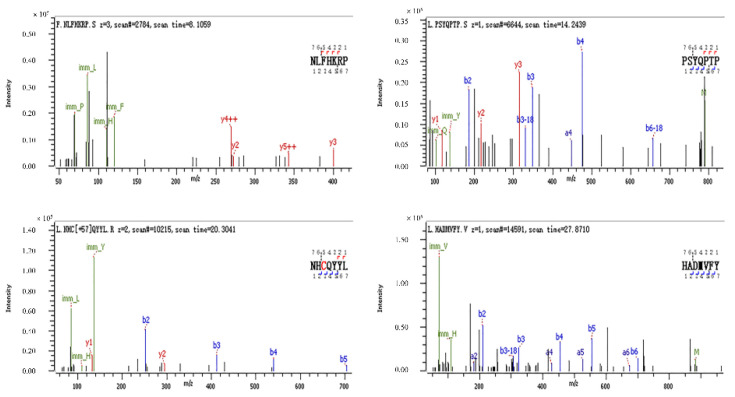
Secondary mass spectra of HADMVFY, NHCQYYL, NLFHKRP and PSYQPTP. The series of b and y ions are fragment ions produced when the peptide breaks at the peptide bond.

**Figure 10 molecules-27-08423-f010:**
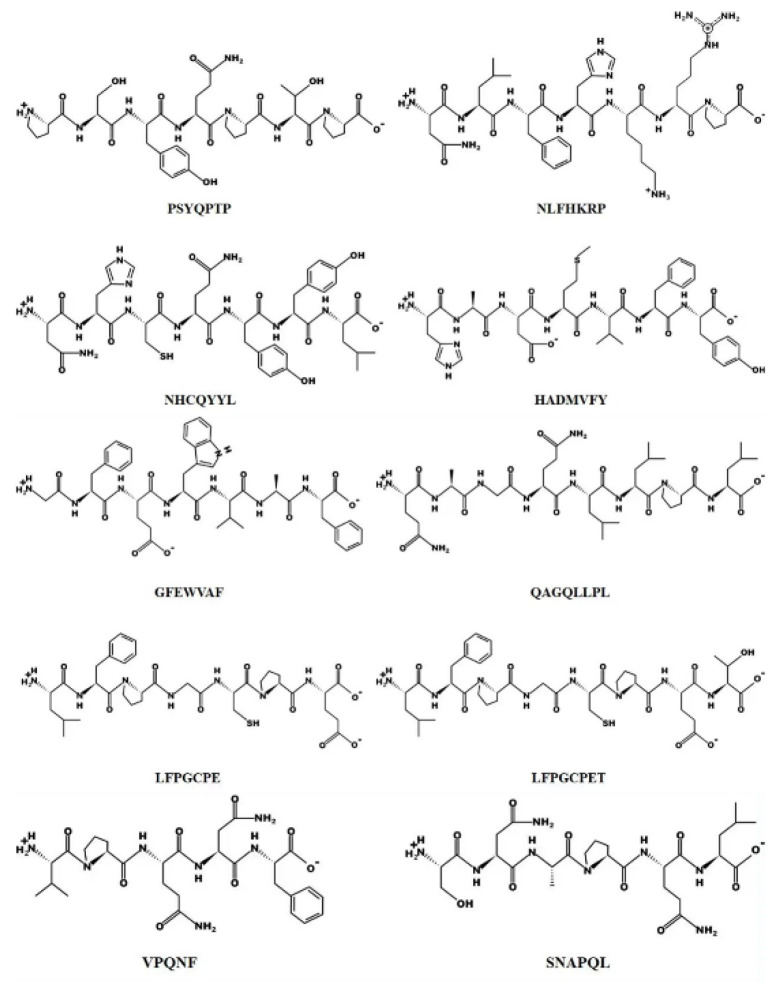
Chemical structures of the ten peptides.

**Figure 11 molecules-27-08423-f011:**
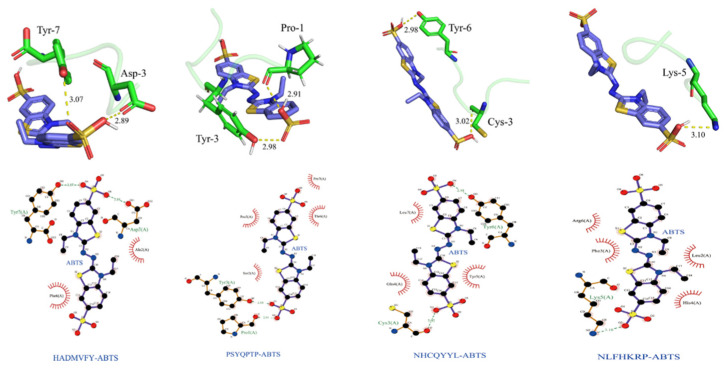
Dominant conformations and interaction models of HADMVFY, NHCQYYL, NLFHKRP and PSYQPTP docked with ABTS.

**Table 1 molecules-27-08423-t001:** Optimal enzymatic conditions for different enzymes.

Type of Enzyme	pH	Temperature (°C)
Alcalase	9.0	55
Flavourzyme	7.5	45
Neutrase	7.0	50
Pepsin	2.0	37
Trypsin	7.0	37

**Table 2 molecules-27-08423-t002:** Parameters a and b at different initial substrate concentration and enzyme concentration.

S_0_ (mg/mL)	E_0_ (mg/mL)	E_0_/S_0_	a (min^−1^)	b	R^2^
40	0.6	0.015	0.660	0.303	0.997
40	0.8	0.020	0.794	0.289	0.998
40	1.0	0.025	0.939	0.299	0.997
40	1.2	0.030	1.085	0.302	0.995
20	0.8	0.040	1.293	0.273	0.998
40	0.8	0.020	0.787	0.235	0.998
60	0.8	0.013	0.662	0.229	0.997
80	0.8	0.010	0.561	0.218	0.998

**Table 3 molecules-27-08423-t003:** Amino acid composition (%) of WPHs * at different hydrolysis times.

AA	Control	30 min	60 min	90 min	120 min	150 min
Asp	10.03	10.58	10.27	10.22	10.09	10.10
Thr	3.66	3.49	3.54	3.62	3.43	3.44
Ser	5.28	4.85	5.33	5.53	5.27	5.35
Glu	23.46	21.71	22.19	22.26	22.44	22.6
Gly	4.46	4.45	4.46	4.38	4.4	4.39
Ala	4.01	4.63	4.46	4.36	4.34	4.32
Cys	0.55	0.23	0.21	0.34	0.23	0.25
Val	4.39	5.07	4.83	4.98	4.74	4.70
Met	0.15	1.34	1.38	1.24	1.57	1.46
Ile	3.65	3.95	3.67	3.59	3.75	3.59
Leu	6.12	7.15	6.96	6.64	6.96	6.89
Tyr	1.72	2.72	2.81	2.5	2.76	2.86
Phe	4.48	4.57	4.43	4.5	4.49	4.49
Lys	3.56	2.7	2.61	2.6	2.55	2.53
His	2.33	2.82	2.75	2.69	2.74	2.85
Arg	14.95	14.56	15.01	14.64	14.97	14.98
Pro	7.19	5.17	5.10	5.91	5.27	5.20
EAA	26.01	28.27	27.42	27.17	27.48	27.1
SCAA	0.70	1.57	1.59	1.58	1.80	1.71
BCAA	14.16	16.17	15.46	15.21	15.45	15.18
DEAA	23.61	23.05	23.57	23.5	24.01	24.06
PCAA	20.84	20.08	20.37	19.93	20.26	20.36
HAA	31.56	33.27	32.26	32.42	32.49	32.05
AAA	6.20	7.30	7.25	6.94	7.43	7.35

* Essential amino acids (EAA) = Lys, Trp, Phe, Met, Thr, Ile, Leu, Val. Sulfur-containing amino acids (SCAA) = Cys, Met. Branched-chain amino acids (BCAA) = Val, Ile, Leu; Amino acids associated with the capacity of donating electron/hydrogen (DEAA) = Glu, Met. Positively charged amino acids (PCAA) = His, Arg, Lys. Hydrophobic amino acids (HAA) = Ala, Pro, Tyr, Val, Met, Ile, Leu, Phe. Aromatic amino acids (AAA) = Phe, Tyr.

**Table 4 molecules-27-08423-t004:** The secondary structure contents of different WIP hydrolysates calculated from FTIR spectra.

Hydrolysis Time (min)	α-Helical (%)	β-Sheet (%)	β-Turn (%)	Random Coil (%)
Control	21.08%	32.92%	45.99%	0.01%
30	3.88%	60.38%	10.83%	24.92%
60	5.61%	62.07%	13.04%	19.28%
90	5.02%	57.18%	21.82%	15.98%
120	0.01%	14.93%	63.28%	21.78%
150	12.97%	21.35%	59.02%	6.66%

**Table 5 molecules-27-08423-t005:** LC-MS/MS identification of the antioxidant peptides.

Peptide Sequence	Mass (Da)	Origin Protein	Location
GFEWVAF	855.406	11S globulin-like	422–428
LFPGCPE	819.372	11S globulin-like	108–114
QAGQLLPL	839.498	glutaredoxin-like	104–111
HADMVFY	883.384	vicilin-like seed storage protein At2g18540	90–96
VPQNF	604.312	type IV inositol polyphosphate 5-phosphatase 9	170–174
PSYQPTP	789.379	11S globulin-like	85–91
NHCQYYL	997.424	2S sulfur-rich seed storage protein 2	57–63
NLFHKRP	911.519	Haloacid dehalogenase-like hydrolase domain-containing protein At2g33255	143–148
LFPGCPET	920.420	11S globulin-like	110–117
SNAPQL	629.326	11S globulin-like	92–97

**Table 6 molecules-27-08423-t006:** Physicochemical properties of antioxidant peptides.

Peptide Sequence	Toxicity	Steric Hindrance	Amphipathicity	Net Hydrogen	Hydrophobicity (kcal/mol)
PSYQPTP	Non-toxin	0.50	0.18	0.71	+9.09
NLFHKRP	Non-toxin	0.53	1.08	1.29	+12.87
NHCQYYL	Non-toxin	0.57	0.39	1.00	+9.16
HADMVFY	Non-toxin	0.59	0.21	0.43	+10.82
GFEWVAF	Non-toxin	0.64	0.18	0.29	+7.21
QAGQLLPL	Non-toxin	0.56	0.31	0.50	+7.48
LFPGCPE	Non-toxin	0.56	0.18	0.14	+9.98
LFPGCPET	Non-toxin	0.56	0.16	0.25	+10.23
VPQNF	Non-toxin	0.64	0.25	0.80	+7.49
SNAPQL	Non-toxin	0.56	0.21	0.83	+9.37

**Table 7 molecules-27-08423-t007:** Molecular docking modeling energy scores and interaction results of nine peptides obtained from WPH.

Number	Peptide Sequence	Length	Binding Energy
1	PSYQPTP	7	−4.423
2	NLFHKRP	7	−4.401
3	NHCQYYL	7	−4.379
4	HADMVFY	7	−4.263
5	GFEWVAF	7	−4.181
6	QAGQLLPL	8	−4.143
7	LFPGCPE	7	−4.094
8	LFPGCPET	8	−4.08
9	VPQNF	5	−3.99
10	SNAPQL	6	−3.592

## Data Availability

Data are contained within the article.

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
