# Peer review of "Purification, Identification and Molecular Docking of Novel Antioxidant Peptides from Walnut (Juglans regia L.) Protein Hydrolysates"

_molecules, 2022, doi:10.3390/molecules27238423_

Round 1
Reviewer 1 Report
Dear Authors,
You present here the results of your work regarding the purification, identification and molecular docking of novel antioxidant peptides from walnut protein hydrolysates. This may become a subject of interest for the readers, but with some corrections/clarifications needed:
1. in the antioxidant assay, there is no reference substance used as antioxidant
2. fig. 6 should appear after its citation in text. Also, correct "rudcing" which appears in this figure. Still here, you should describe what are the letters a-f used for
3. in the title, Juglans regia L. should be written in italic. Also, the terms "in vitro" which appear in several places in the manuscript
4. for the affirmation in lines 31-32, regarding the toxicity of some synthetic antioxidants, you need to add a reference. Which synthetic antioxidant presents toxicity and what type of toxicity?
5. many paragraphs should be rephrased, due to their lack of clarity or meaning, such as lines 47, 82, 84, 89
6. correct "furfure use" in line 94
7. there are some different fonts or size letters used, such in line 97
8. what do you mean by alpha-NH2 in line 100, an aminoacid?
9. what is the triple bound between C and O, in line 420?
10. reduce the size of Figure 1. Also, include the abbreviations in figure legend, because a figure should stand by itself, should be easily understood without even reading the explanations given
Reviewer 2 Report
This work present novel antioxidant peptide from walnut protein. This manuscript (MS) need changes which are mentioned as following
Paper format: The paper is in molecules required format. Except for equations throughout the MS.
Title: Okay
Abstract: The abstract need revision in terms of including significant values among different data. Avoid theoretical approach in abstract, just show what results are saying. Keep to the point and address novelty of the work. The abstract should always be concise and informative. The arguments of why your study is important not making any sense. Extensive revision is required in abstract, as the present sentences sounds noisy during reading. Overall the abstract is not informative enough and need to show the actual picture of the work. Authors are supposed to please indicate the numerical values with significant difference in abstract. Keep abstract minimum with covering all of your data.
Keywords: Not according to manuscript. Please change it to some structural features study
Introduction: some shortcoming are below
1. Line 33, It should be plant instead of food. Change it
2. Line 34 ‘The public is also increasingly inclined to choose natural antioxidants’. The sentence are broadly used in recent time, I am surprised that till date no one provide that where such study has been conducted? Who conducted survey in general public? Modify this sentence please or give reference here.
3. Line 37 why Torreya grandis is in italic form? Put English name in it.
4. Line 53 Walnuts (Juglans regia L.) is not pecan, pecan are Carya illinoinensis, this is totally wrong, change it please.
5. Novelty is missing in introduction, add few sentence to justify your work, with proper aim and objectives.
1. What is the safety and environment concern for such methods? elaborate
2. Please add/indicate and compare your work with pervious published paper.
3. Please cite the following latest papers
https://link.springer.com/article/10.1007/s10068-019-00590-z
https://pubmed.ncbi.nlm.nih.gov/36193000/
https://www.ncbi.nlm.nih.gov/pmc/articles/PMC7143977/
Methods and materials:
1. These experiments should be divide in three parts i.e. structural study of hydrolysate, physiochemical properties and then functional properties. Go with three parts.
2. No solubility data has been provided for walnut protein, how to believe the enzymatic reaction between protein and enzyme? What best pH for reaction?
3. Line 89 what was the pH? Adjust with buffer? Mention here
4. 2.4 section of experiments is missing, add details of the experiments.
5. Check format of the equation equation-1.
6. Check all of the equations format and numbering
Results and discussion:
1. I have through the results and discussion, discussion is fair enough but figures in this study are very poor, hardly to understand the point which authors are trying to address. For example see figures 2 & 3 and same figures 8 & 9, I did not understand what authors are looking in these data. All of the figures are unreadable.
2. Total ion chromatogram figure is not acceptable, tabulate these data please.
Conclusion: Change the conclusion as per changes in discussion, add more of application for this research for common reader.
Round 2
Reviewer 1 Report
Dear Authors,
Thank you for considering my comments and suggestions.
Reviewer 2 Report
Suggestions made by authors are fine and they did fine job in revision. Accept for publication